# Accuracy of ICD-9 codes in identifying patients with peptic ulcer and gastrointestinal hemorrhage in the regional healthcare administrative database of Umbria

Massimiliano Orso[1,2], Iosief Abraha[1,3]*, Anna Mengoni[2], Fabrizio Taborchi[4], Marcello De Giorgi[5], David Franchini[5], Paolo Eusebi[1], Anna Julia Heymann[6], Alessandro Montedori[1], Giuseppe Ambrosio[2], Francesco Cozzolino[1,2]

1 Health Planning Service, Regional Health Authority of Umbria, Perugia, Italy, 2 Division of Cardiology, Santa Maria della Misericordia Hospital, University of Perugia School of Medicine, Perugia, Italy, 3 Centro Regionale Sangue, Servizio Immunotrasfusionale, Azienda Ospedaliera di Perugia, Perugia, Italy, 4 Gastroenterologia, Azienda Ospedaliera di Perugia, Perugia, Italy, 5 Health ICT Service, Regional Health Authority of Umbria, Perugia, Italy, 6 Istituto Zooprofilattico Sperimentale dell'Umbria e delle Marche, Perugia, Italy

* iosief_a@yahoo.it

**Data Availability Statement:** All relevant data are within the manuscript and its Supporting Information files.

## Abstract

### Background

Peptic ulcer is a widespread disease, frequently complicated by perforation and bleeding. Administrative databases are useful tool to perform epidemiological and drug utilization studies, but they need a validation process based on a comparison with the original data contained in the medical charts. Our aim was to evaluate the accuracy of the ICD-9 codes in identifying patients with peptic ulcer and gastrointestinal hemorrhage in the regional administrative database of Umbria.

### Methods

The index test of our study was the hospital discharge abstract database of the Umbria region (Italy), while the reference standard was the clinical information collected in the medical charts. The study population were adult patients with a hospital discharge for peptic ulcer or gastrointestinal hemorrhage in the period 2012–2014. A random sample of cases and non-cases was selected and the corresponding medical charts were reviewed. Cases of peptic ulcer were confirmed based on endoscopy, radiology, and surgery, while adjudication of gastrointestinal hemorrhage was based on presence of hematemesis, melena, and rectal bleeding.

### Results

Overall, we reviewed 445 clinical charts of cases and 80 clinical charts of non-cases. The diagnostic accuracy results were: code 531 (gastric ulcer), sensitivity and NPV 98%, specificity 88%, and PPV 91%; code 532 (duodenal ulcer), sensitivity and NPV 100%, specificity

**Funding:** This study was developed within the Data-Value Project ("Progetto Data-Value: valorizzazione del dato sanitario regionale per la Ricerca dei Servizi Sanitari (Health Services Research)" – D.G.R. No 1798 of 29/12/2014) supported by funding from the Regional Health Authority of Umbria. The funder had no role in study design, data collection and analysis, decision to publish, or preparation of the manuscript.

**Competing interests:** The authors have declared that no competing interests exist.

and PPV 98%; code 534 (gastrojejunal ulcer), sensitivity and NPV 100%, specificity 70%, and PPV 45%; code 578 (gastrointestinal hemorrhage), sensitivity 96%, specificity 90%, PPV and NPV 94%.

## Conclusions

Our results showed a high level of diagnostic accuracy for most of the codes considered. The ICD-9 code 534 of gastrojejunal ulcer had a lower level of specificity and PPV due to false positives, being mainly misclassifications for coding errors. These validated codes can be used for future epidemiological studies and for health services research.

## Introduction

Administrative healthcare databases collect a great amount of demographic data, drug prescriptions, diagnostic and therapeutic procedures. The information contained in these databases are not collected with a research purpose and, to be used for this scope, they should be previously validated.

Peptic ulcer is a common disease with a worldwide prevalence of 5–10% and an incidence of 0.1–0.3% per year [1]. The most frequent complications of peptic ulcer disease are perforation and bleeding. A systematic review reported an annual incidence of hemorrhage in the general population ranging from 0.02 to 0.06%, and an annual incidence of perforation ranging from 0.004 to 0.014% [2]. Traditionally risk factors for peptic ulcer disease involve a hypersecretory acid environment, dietary factors, and stress, while detection of Helicobacter pylori infection, frequent use of nonsteroidal anti-inflammatory drugs (NSAIDs), alcohol consumption, and smoking abuse have modified the etiology of this disease.

The frequent use of NSAIDs and anticoagulant drugs for the treatment of cardiovascular and cerebrovascular diseases represents the main cause of gastrointestinal bleeding. The present study is part of two other validation studies of cardiovascular [3] and cerebrovascular diseases [4].

The objective of this study was to assess the accuracy of the ICD-9 codes in identifying patients with peptic ulcer and gastrointestinal hemorrhage in the administrative database of the Regional Health Authority of Umbria.

## Materials and methods

### Setting and data source

**Administrative database.** The index test considered in the present study was the hospital discharge abstract database of the Umbria Region (Italy). This database collects data on all hospital admissions of all 890,000 residents, and contains information on personal demographics, admitting and discharge date, vital status, ICD-9 codes of primary and secondary diagnoses, diagnostic tests, medications, and surgical procedures. Each resident has a unique personal identifier within the database that allows a record linkage with other databases, such as the drug prescription database.

**Source population.** We considered all the residents in the Umbria Region > 18 years discharged from seven hospitals (Perugia, Terni, Foligno, Città di Castello, Orvieto, Gubbio-Gualdo Tadino, Spoleto) between 2012 and 2014 with a diagnosis of peptic ulcer or

gastrointestinal hemorrhage. We excluded residents hospitalised outside the regional territory of Umbria.

**Case selection and sampling method.** The methodology of this study for case selection and sampling method is based on that described on our research protocol for cardiovascular and cerebrovascular diseases [5]. Through a simple randomization method using SAS 9.4 we selected from the administrative database of Umbria four cohorts of "cases", that is incident patients with a diagnosis of peptic ulcer and gastrointestinal hemorrhage between 2012 and 2014 having in the discharge abstract the ICD-9 codes located in primary position of gastric ulcer (ICD-9 code 531), duodenal ulcer (code 532), gastrojejunal ulcer (codes 534), gastrointestinal hemorrhage (codes 578). The ICD-9 code 533 "Peptic ulcer, site unspecified" was initially considered for validation, but we found only five cases with this diagnosis in primary position and we decided to exclude it from the final analysis. From our cohorts we excluded patients discharged with the same diagnosis from 2007 to 2011.

From original cohorts we extracted a random sample of 130 cases for the codes 531, 532, and 578, while for the code 534 we considered all the patients discharged. In addition, we selected a cohort of "non-cases", i.e. patients who had been discharged in the same period in a gastroenterology ward with a diagnosis other than peptic ulcer or gastrointestinal hemorrhage, from which we extracted a random sample of 80 patients. This sample of non-cases was used as control group for each of the four diseases.

## Chart abstraction and case ascertainment

We retrieved the following data from the medical charts of cases and non-cases: clinical chart number, date of birth, gender, dates of hospital admission and discharge, hospital discharge procedure, primary and secondary diagnoses, medical history, any diagnostic procedure and treatment that contributed to the diagnosis of the disease.

Clinical charts were reviewed by physicians previously trained in data extraction. We performed a pilot phase in which the reviewers independently examined 25 clinical charts, with a level of agreement very high (k> 0.88). To achieve a higher level of agreement the working group discussed about the cases of disagreement that were solved by the judgement of a third reviewer (GA). Data extraction was performed using predetermined data extraction sheets.

## Validation criteria

To validate the ICD-9 codes for peptic ulcer we considered endoscopy, radiology, and surgery, while to validate gastrointestinal hemorrhage we considered the occurrence of hematemesis, melena, and rectal bleeding.

## Statistical analysis

We calculated a sample of 125 cases and 80 non-cases in order to obtain an expected positive predictive value (PPV) of 73% (estimated median from available published studies [6–11]) and a negative predictive value (NPV) of 90% (our assumption in absence of published evidence) with a maximum width of the 95% CI of 16% according to exact calculation [12].

For each ICD-9 code, we calculated sensitivity, specificity, PPV and NPV, along with their corresponding 95% CI.

## Reporting

Quality of reporting was guaranteed following the Standards for Reporting Diagnostic Accuracy (STARD) criteria [13] (S1 Table).

### Ethics statement

Ethics approval has been obtained from the Regional Ethics Committee of Umbria (CEAS), registry No 2695/15 of 16/12/2015.

## Results

A random sample of 130 medical charts for each cohort of cases, and 80 medical charts from the cohort of non-cases was selected. The total number of clinical charts reviewed for cases was 445: 128 each for gastric (ICD-9 code 531) and duodenal ulcer (ICD-9 code 532), 62 for gastrojejunal ulcer (ICD-9 code 534), and 127 for gastrointestinal hemorrhage (ICD-9 code 578). For gastrojejunal ulcer, we considered all the available hospital admissions in the period 2012–2014. In the meanwhile, we selected 80 clinical charts for non-cases. For each ICD-9 code, characteristics of the patients are described in Tables 1–4.

**Table 1. Characteristics of patients with gastric ulcer.**

| **Gastric ulcer** | |
|---|---|
| **Incident cases** (N medical charts reviewed) | 128 |
| **ICD-9 code, N (%)** | |
| 531 Gastric ulcer | 128 (100%) |
| 531.0 Acute with hemorrhage | 78 (61%) |
| 531.1 Acute with perforation | 15 (12%) |
| 531.2 Acute with hemorrhage and perforation | 2 (2%) |
| 531.3 Acute without mention of hemorrhage or perforation | 21 (16%) |
| 531.4 Chronic or unspecified with hemorrhage | 4 (3%) |
| 531.5 Chronic or unspecified with perforation | - |
| 531.6 Chronic or unspecified with hemorrhage and perforation | - |
| 531.7 Chronic without mention of hemorrhage or perforation | 2 (2%) |
| 531.9 Unspecified as acute or chronic, without mention of hemorrhage or perforation | 6 (5%) |
| **Sex, N (%)** | |
| Male | 77 (60%) |
| Female | 51 (40%) |
| **Age, N (%)** | |
| *< 60* | 25 (20%) |
| *60–79* | 55 (43%) |
| *≥ 80* | 48 (38%) |
| **Instrumental examinations, N (%)** | |
| Gastroscopy | 117 (91%) |
| Abdominal ultrasound | 43 (34%) |
| Abdominal CT | 14 (11%) |
| Abdominal x-ray | 10 (8%) |
| **Histological documentation, N (%)** | |
| Biopsy from gastroscopy | 55 (43%) |
| Biopsy from surgery | 9 (7%) |
| **Surgical procedures, N (%)** | |
| Gastrectomy | 6 (5%) |
| Other surgical procedures | 7 (5%) |
| **Laboratory analyses, N (%)** | |
| Haemoglobin levels | 122 (95%) |

**Table 2. Characteristics of patients with duodenal ulcer.**

| Duodenal ulcer | |
|---|---|
| **Incident cases** (N medical charts reviewed) | 128 |
| **ICD-9 code, N (%)** | |
| 532 Duodenal ulcer | 128 (100%) |
| 532.0 Acute with hemorrhage | 70 (55%) |
| 532.1 Acute with perforation | 6 (5%) |
| 532.2 Acute with hemorrhage and perforation | 4 (3%) |
| 532.3 Acute without mention of hemorrhage or perforation | 20 (16%) |
| 532.4 Chronic or unspecified with hemorrhage | 11 (9%) |
| 532.5 Chronic or unspecified with perforation | 6 (5%) |
| 532.6 Chronic or unspecified with hemorrhage and perforation | 1 (1%) |
| 532.7 Chronic without mention of hemorrhage or perforation | 1 (1%) |
| 532.9 Unspecified as acute or chronic, without mention of hemorrhage or perforation | 9 (7%) |
| **Sex** | |
| Male | 80 (63%) |
| Female | 48 (38%) |
| **Age, N (%)** | |
| < 60 | 42 (33%) |
| 60–79 | 43 (34%) |
| ≥ 80 | 43 (34%) |
| **Instrumental examinations, N (%)** | |
| Gastroscopy | 115 (90%) |
| Abdominal ultrasound | 41 (32%) |
| Abdominal CT | 13 (10%) |
| Abdominal x-ray | 14 (11%) |
| **Histological documentation, N (%)** | |
| Biopsy from gastroscopy | 37 (29%) |
| Biopsy from surgery | 7 (5%) |
| **Surgical procedures, N (%)** | |
| Gastrectomy | 5 (4%) |
| Other surgical procedures | 13 (10%) |
| **Laboratory analyses, N (%)** | |
| Haemoglobin levels | 122 (95%) |

A minimal anonymized dataset is provided as an additional support information file (S1 Dataset).

The cross tabulation reporting the index test and reference standard results is reported in Table 5.

## Gastric ulcer

We identified 358 patients having the ICD-9 code 531 in primary position between 2012 and 2014. From this cohort, we extracted a sample of 130 cases, of these 128 were analysed (two clinical charts were not available).

The general characteristics of the patients with gastric ulcer are described in Table 1. Most of patients were males (60%) and > 60 years (80%).

**Table 3. Characteristics of patients with gastrojejunal ulcer.**

| Gastrojejunal ulcer | |
|---|---|
| **Incident cases** (N medical charts reviewed) | 62 |
| **ICD-9 code, N (%)** | |
| 534 Gastrojejunal ulcer | 62 (100%) |
| 534.0 Acute with hemorrhage | 47 (76%) |
| 534.1 Acute with perforation | 4 (6%) |
| 534.2 Acute with hemorrhage and perforation | - |
| 534.3 Acute without mention of hemorrhage or perforation | 3 (5%) |
| 534.4 Chronic or unspecified with hemorrhage | 3 (5%) |
| 534.5 Chronic or unspecified with perforation | - |
| 534.6 Chronic or unspecified with hemorrhage and perforation | 2 (3%) |
| 534.7 Chronic without mention of hemorrhage or perforation | 1 (2%) |
| 534.9 Unspecified as acute or chronic, without mention of hemorrhage or perforation | 2 (3%) |
| **Sex** | |
| Male | 34 (55%) |
| Female | 28 (45%) |
| **Age, N (%)** | |
| < 60 | 8 (13%) |
| 60–79 | 26 (42%) |
| ≥ 80 | 28 (45%) |
| **Instrumental examinations, N (%)** | |
| Gastroscopy | 56 (90%) |
| Abdominal ultrasound | 14 (23%) |
| Abdominal CT | 8 (13%) |
| Abdominal x-ray | 4 (6%) |
| **Histological documentation, N (%)** | |
| Biopsy from gastroscopy | 21 (34%) |
| Biopsy from surgery | 2 (3%) |
| **Surgical procedures, N (%)** | |
| Gastrectomy | 2 (3%) |
| Other surgical procedures | 3 (5%) |
| **Laboratory analyses, N (%)** | |
| Haemoglobin levels | 61 (98%) |

Gastroscopy was the diagnostic test mostly performed (91%), followed by abdominal ultrasound (34%). We found histological documentation from biopsy in 43% of clinical charts, while the surgical procedures occurred in 10% of patients.

The diagnostic accuracy measures derived from the cross tabulation (Table 5) are: sensitivity 98% (95% CI: 94%–100%), specificity 88% (95% CI: 79%–94%), PPV 91% (95% CI: 85%–96%), and NPV 98% (95% CI: 91%–100%). Misclassification of cases and non-cases is described in Table 6.

The false positives (n. 11) were due to coding errors, and gastroscopy or histology by biopsy negative for gastric ulcer, while the false negatives (n.2) were patients with gastric ulcer diagnosed by gastroscopy (code 531 in secondary position).

## Duodenal ulcer

We identified 351 cases having the ICD-9 code 532 in primary position between 2012 and 2014. From this cohort, we extracted a sample of 130 cases, of these 128 were analysed (two clinical charts were not available).

**Table 4. Characteristics of patients with gastrointestinal hemorrhage.**

| Gastrointestinal hemorrhage | |
|---|---|
| **Incident cases** (N medical charts reviewed) | 127 |
| **ICD-9 code, N (%)** | |
| 578 Gastrointestinal hemorrhage | 127 (100%) |
| 578.0 Hematemesis | 13 (10%) |
| 578.1 Blood in stool | 70 (55%) |
| 578.9 Hemorrhage of gastrointestinal tract, unspecified | 44 (35%) |
| **Sex** | |
| Male | 66 (52%) |
| Female | 61 (48%) |
| **Age, N (%)** | |
| < 60 | 15 (12%) |
| 60–79 | 57 (45%) |
| ≥ 80 | 55 (43%) |
| **Instrumental examinations, N (%)** | |
| Gastroscopy | 58 (46%) |
| Colonoscopy | 62 (49%) |
| Abdominal ultrasound | 29 (23%) |
| Abdominal CT | 14 (11%) |
| Abdominal x-ray | 4 (3%) |
| **Histological documentation, N (%)** | |
| Biopsy from gastroscopy | 10 (8%) |
| Biopsy from colonscopy | 15 (12%) |
| **Laboratory analyses, N (%)** | |
| Haemoglobin levels | 124 (98%) |
| **Deaths, N (%)** | |
| Patients deceased during hospital admission | 11 (9%) |

The general characteristics of the patients with duodenal ulcer are described in Table 2. Most of patients were males (63%), while patients were equally distributed between the three age classes considered.

Gastroscopy was the diagnostic test mostly performed (90%), followed by abdominal ultrasound (32%). We found histological documentation from biopsy in 29% of clinical charts, while the surgical procedures occurred in 14% of patients.

The diagnostic accuracy measures derived from the cross tabulation (Table 5) are: sensitivity 100% (95% CI: 97%–100%), specificity 98% (95% CI: 92%–100%), PPV 98% (95% CI: 95%–100%), and NPV 100% (95% CI: 96%–100%). Misclassification of cases and non-cases is described in Table 6.

The false positives (n. 2) were due to duodenal ulcer not found by gastroscopy.

## Gastrojejunal ulcer

We identified 63 overall cases having the ICD-9 code 534 in primary position between 2012 and 2014, and of these 62 were analysed (one clinical chart was not available).

The general characteristics of the patients with gastrojejunal ulcer are described in Table 3. Most of patients were males (55%) and > 60 years (87%).

**Table 5. Cross tabulation of the index test (ICD-9-CM code) for the results of the reference standard (medical chart).**

| | True Positive | False Positive | True Negative | False Negative |
|---|---|---|---|---|
| 531 Gastric ulcer | 117 | 11 | 78 | 2 |
| 532 Duodenal ulcer | 126 | 2 | 80 | 0 |
| 534 Gastrojejunal ulcer | 28 | 34 | 80 | 0 |
| 578 Gastrointestinal hemorrhage | 119 | 8 | 75 | 5 |

Gastroscopy was the diagnostic test mostly performed (90%), followed by abdominal ultrasound (23%). We found histological documentation from biopsy in 34% of clinical charts, while the surgical procedures occurred in 8% of patients.

The diagnostic accuracy measures derived from the cross tabulation (Table 5) are: sensitivity 100% (95% CI: 88%–100%), specificity 70% (95% CI: 61%–78%), PPV 45% (95% CI: 33%–58%), and NPV 100% (95% CI: 96%–100%). Misclassification of cases and non-cases is described in Table 6.

The false positives (n. 34) were mostly due to coding errors (n. 26), and to gastroscopy negative for gastrojejunal ulcer or not reported (n. 8).

## Gastrointestinal hemorrhage

We identified 947 patients having the ICD-9 code 578 in primary position between 2012 and 2014. From this cohort, we extracted a sample of 130 cases, of these 127 were analysed (three clinical charts were not available).

The general characteristics of the patients with gastrointestinal hemorrhage are described in Table 4. Patients were equally distributed between sex, while most of patients were > 60 years (88%).

Gastroscopy and coloscopy were the diagnostic tests mostly performed (46% and 49% respectively), followed by abdominal ultrasound (23%). We found that almost all patients (98%) had haemoglobin levels from laboratory analysis. Nine percent of patients died during hospital stay.

The diagnostic accuracy measures derived from the cross tabulation (Table 5) are: sensitivity 96% (95% CI: 91%–99%), specificity 90% (95% CI: 82%–96%), PPV 94% (95% CI: 88%–97%), and NPV 94% (95% CI: 86%–98%). Misclassification of cases and non-cases is described in Table 6.

**Table 6. Reasons for incorrect identification of cases and controls.**

| | 531 Gastric ulcer | 532 Duodenal ulcer | 534 Gastrojejunal ulcer | 578 Gastrointestinal hemorrhage |
|---|---|---|---|---|
| **FALSE POSITIVES** | - Misclassifications (3 gastrojejunal ulcers and 1 duodenal ulcer): n.4;<br>- Gastric ulcer not found by gastroscopy: n. 6;<br>- Gastroscopy report not found in the clinical chart and histology from biopsy negative for gastric ulcer: n. 1. | - Duodenal ulcer not found by gastroscopy: n. 2. | - Misclassifications (19 gastric ulcers, 6 duodenal ulcers, 1 occlusion of cerebral arteries): n. 26;<br>- Gastrojejunal ulcer not found by gastroscopy: n. 7;<br>- Gastroscopy report not found in the clinical chart: n. 1. | - Blood in stool not found: n. 2;<br>- Misclassifications (2 gastric ulcers, 1 duodenal ulcer, 2 gastrojejunal ulcers): n. 5;<br>- Hemorrhage of gastrointestinal tract not found: n. 1. |
| **FALSE NEGATIVES** | Patients with gastric ulcer diagnosed by gastroscopy (code 531 in secondary position): n. 2. | None | None | - Patients with blood in stool: n. 4;<br>- Patient with hematemesis: n. 1. |

The false positives (n. 8) were due to coding errors (n. 5), and blood in stool or hemorrhage of gastrointestinal tract not found (n. 3), while the false negatives (n. 5) were patients having blood in stool or hematemesis.

## Discussion

The present study is one of the few in Italy and the first in Umbria Region validating the ICD-9 codes related to peptic ulcer and gastrointestinal hemorrhage using clinical charts as a reference standard. We performed a literature search to find studies validating the same diseases in Italy or worldwide. We did not find any systematic review on this topic, but only primary diagnostic accuracy studies validating the same ICD-9 codes of our study, with some differences on study design and ICD-9 sub-codes considered.

The results of our study in terms of PPV are in line with those found in other validation studies considering clinical charts as the reference standard.

Cattaruzzi et al. [7] performed a validation study in the Italian region of Friuli–Venezia Giulia, identifying patients with upper gastrointestinal bleeding (UGIB) and perforation to estimate the risk of hospitalization associated with intake of nonsteroidal antiinflammatory drugs (NSAIDs) and other drugs. They considered the same ICD-9 codes of our study (peptic ulcer and gastrointestinal bleeding), limited to the sub-codes of hemorrhage or perforation. The overall PPV for the code 531 for a confirmed site of UGIB was 89%, 532 code 83%, 534 code 46%, and 578 code ranging from 59% to 70%.

Another more recent Italian validation study having the same objectives of the previous study [7] was carried out by Pisa et al. [9]. The PPV results were: 531 code 66%, 532 code 92%, 534 code 33%, and 578 code 33–51%. Compared to Pisa [9] results, our study found a higher PPV value for the codes 531 and 578.

In addition, we retrieved other three international studies on this topic [6, 10, 11]. Raiford and colleagues [10] calculated the PPV of ICD-9 codes used to identify cases of complicated peptic ulcer disease from the Saskatchewan Hospital automated database. The overall PPV for the code 531 for a confirmed site of UGIB was 83%, 532 code 81%, and 578 code 84–88%; no case was detected for 534 code.

Another study conducted in USA [6] evaluated the PPV of ICD-9 codes for cases of peptic ulcers and upper gastrointestinal bleeding documented in eight large health maintenance organizations (HMOs) databases. The authors evaluated the codes 531 and 534 together. The PPVs were 77% for the code 532 of duodenal ulcer, 76% for gastric/gastrojejunal ulcer (codes 531+534), and 7% for gastrointestinal hemorrhage. The PPV for the code 578 was very lower compared to other studies [7, 10], probably due to more stringent criteria for case definition of upper gastrointestinal hemorrhage (from gastric or duodenal ulcer, hemorrhagic gastritis, or duodenitis) confirmed by surgery, endoscopy, X-ray, or autopsy.

The last study found was that of Viborg et al. [11] developed in Denmark. This study was aimed to validate the ICD-10 codes of peptic ulcer in the Danish National Patient Registry (DNPR) by estimating PPVs only for gastric and duodenal ulcer diagnoses. The PPV of gastric ulcer diagnosis (ICD-10 code K25) in DNPR was 90%, and for duodenal ulcer (ICD-10 code K26) was 94%.

All the studies found assessed only the PPV, not considering a control group of patients without a diagnosis of peptic ulcer or gastrointestinal hemorrhage. Instead, in order to estimate sensitivity and specificity, in absence of a disease registry for peptic ulcers and gastrointestinal hemorrhage that constitutes the real prevalence of the diseases, we chose to consider a sample of "non-cases", i.e. patients who had been discharged in the same period in a

gastroenterology ward with a diagnosis other than peptic ulcer or gastrointestinal hemorrhage, to individuate possible false negatives.

Another consideration is about the lower value of PPV found in our study for the code 534 compared to the other codes, mostly due to several coding errors. However, this low PPV is comparable with those reported in other above-mentioned studies [7, 9].

Regarding the generalizability of our study, we want to highlight that, in general, validation studies of administrative databases are context-specific due to differences that may exist in demographics, disease prevalence, and standards of care among different contexts, and thus our results can confidently be applied only to the regional setting of Umbria. However, our methodology could be replicated in other regional or national settings in order to identify possible differences in diagnostic accuracy measures results.

## Strengths and limitations

A strength of our study is that we used medical charts as the reference standard for case ascertainment of peptic ulcer and gastrointestinal hemorrhage.

Our methodology derives from a published protocol on cardiovascular and cerebrovascular diseases. Quality of reporting was ensured following the STARD 2015 criteria [13] for diagnostic accuracy studies. Finally, we considered detailed and explicit criteria for case ascertainment, and the data extraction from clinical charts was performed in duplicate and independent way.

We acknowledge that a potential limitation of our study is that we evaluated the accuracy of ICD-9 codes located only in primary position. We chose to limit our analysis only to the codes in primary position because, according to the Italian legislation, the primary diagnosis constitutes the main cause of the need for treatment and/or diagnostic tests, and is mainly responsible for the use of resources.

Another possible limitation of the present study concerns the generalizability of our results in other geographical settings with different demographic characteristics and disease prevalence.

## Conclusion

In this study, we validated the ICD-9 diagnostic codes for peptic ulcer and gastrointestinal hemorrhage using the Regional Healthcare administrative database of Umbria. Most of the ICD-9 codes considered (531, 532, and 578) showed a high level for all the diagnostic accuracy measures. The ICD-9 code 534 had a very high level of sensitivity and NPV, but lower levels of specificity and PPV due to false positives, mainly for coding errors.

According to our results, the validated codes for peptic ulcer and gastrointestinal hemorrhage could be used in future studies evaluating epidemiological and clinical research on health services.

## Supporting information

**S1 Table. STARD-2015-checklist.**
(DOCX)

**S1 Dataset. Minimal anonymized dataset.**
(PDF)

## Author Contributions

**Conceptualization:** Massimiliano Orso, Iosief Abraha, Alessandro Montedori, Giuseppe Ambrosio, Francesco Cozzolino.

**Data curation:** Massimiliano Orso, Iosief Abraha, Francesco Cozzolino.

**Formal analysis:** Massimiliano Orso, Iosief Abraha, Paolo Eusebi, Francesco Cozzolino.

**Funding acquisition:** Alessandro Montedori, Giuseppe Ambrosio.

**Investigation:** Massimiliano Orso, Iosief Abraha, Anna Mengoni, Francesco Cozzolino.

**Methodology:** Massimiliano Orso, Iosief Abraha, Francesco Cozzolino.

**Project administration:** Iosief Abraha, Alessandro Montedori, Giuseppe Ambrosio.

**Resources:** Iosief Abraha, Marcello De Giorgi, David Franchini, Anna Julia Heymann, Alessandro Montedori.

**Software:** Alessandro Montedori, Giuseppe Ambrosio.

**Supervision:** Iosief Abraha, Alessandro Montedori, Giuseppe Ambrosio.

**Validation:** Massimiliano Orso, Iosief Abraha, Fabrizio Taborchi, Alessandro Montedori, Francesco Cozzolino.

**Visualization:** Massimiliano Orso, Iosief Abraha, Anna Julia Heymann, Francesco Cozzolino.

**Writing – original draft:** Massimiliano Orso, Iosief Abraha, Francesco Cozzolino.

**Writing – review & editing:** Massimiliano Orso, Iosief Abraha, Anna Mengoni, Fabrizio Taborchi, Marcello De Giorgi, David Franchini, Paolo Eusebi, Anna Julia Heymann, Alessandro Montedori, Giuseppe Ambrosio, Francesco Cozzolino.

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
