## [Decision Letter · Decision Letter 0]

15 May 2020

PONE-D-20-09372

Accuracy of ICD-9 codes in identifying patients with peptic ulcer and gastrointestinal hemorrhage in the regional healthcare administrative database of Umbria.

PLOS ONE

Dear Dr. Abraha,

Thank you for submitting your manuscript to PLOS ONE. After careful consideration, we feel that it has merit but does not fully meet PLOS ONE’s publication criteria as it currently stands. Therefore, we invite you to submit a revised version of the manuscript that addresses the points raised during the review process.

The reviewers have provided useful comments to improve the manuscript.

We would appreciate receiving your revised manuscript by Jun 29 2020 11:59PM. To enhance the reproducibility of your results, we recommend that if applicable you deposit your laboratory protocols in protocols.io, where a protocol can be assigned its own identifier (DOI) such that it can be cited independently in the future. For instructions see: http://journals.plos.org/plosone/s/submission-guidelines#loc-laboratory-protocols

We look forward to receiving your revised manuscript.

Kind regards,

Gianni Virgili

Academic Editor

PLOS ONE

Journal Requirements:

Reviewers' comments:

Reviewer's Responses to Questions

**Comments to the Author**

1. Is the manuscript technically sound, and do the data support the conclusions?

Reviewer #1: Yes

Reviewer #2: Yes

2. Has the statistical analysis been performed appropriately and rigorously? 

Reviewer #1: Yes

Reviewer #2: Yes

3. Have the authors made all data underlying the findings in their manuscript fully available?

Reviewer #1: No

Reviewer #2: Yes

4. Is the manuscript presented in an intelligible fashion and written in standard English?

Reviewer #1: Yes

Reviewer #2: Yes

5. Review Comments to the Author

Reviewer #1: In this study authors carried out a validation study by using the Regional Healthcare administrative database of Umbria. As mentioned in the manuscript authors have published other article aimed to validate cardiovascular and cerebrovascular diseases. The article is well written, understandable and well structured. However, I have some comments that authors should take into account:

1) In the method section, the authors declare that they used ICD9 included in the primary position of the hospital discharge database. I am aware that sometime in second position can be included only a few information, but I was wondering why authors did not include also this codes in the validation (i.e., at least for most important event such as gastric haemorrhage).

2) In the discussion authors declare that other studies have been performed in Italy with the same purpose. Can authors better specify the difference among the other two studies and their one? In addition, can authors specify what their paper adds to the Italian literature on this topic?

3) I partly agree with authors’s sentence “….validation studies of administrative database are context-specific, and thus our results can be applied only to the regional setting of Umbria”. What about the use/adaptation of the same codes in other Italian region? Do authors suppose a different diagnosis procedures/codes used in other Italian regions? If yes, why?

Reviewer #2: Dear authors,

Your manuscript gives interesting informations on the accuracy of ICD-9 codes in identifying patients with peptic ulcer and gastrointestinal haemorrhage in the regional healthcare administrative database of Umbria and the validated codes will be useful for future epidemiological studies on health services.

However, there are certain points that need to be clarified or better discussed:

1) In the "setting and datasource paragraph", no information on medical charts were presented. Please explain the origin of these medical charts. From which hospital did you retrive these data? Is there a data warehouse collecting these informations?

2) Despite the great effort in analysing sensitivity and specificity of ICD-9codes, the authors measure those values only on the percentage of 80 “non cases”. Do you know the “real prevalence” of patients with these diseases in the Umbria region between 2012 and 2014? Is there a pathology registry in Umbria to compare the  prevalence of diseases found using administrative data with the “real prevalence” of these diseases in an already validated registry? If these data cannot be recovered (or are inexistent) please discuss in the discussion section.

3) Have you performed any sensitivity analyses considering not only ICD-9 CM codes in the primary position but also using the secondary positions? If not, please  discuss further the motivations in using only primary position codes in the limitations of the study.

4) Please define abbreviations upon first appearance in the text. PPV and NPV abbreviation was firstly used in the "statistical analysis” section but the explanation is reported more than once in the “results” section.

6. PLOS authors have the option to publish the peer review history of their article (what does this mean?). If published, this will include your full peer review and any attached files.

Reviewer #1: No

Reviewer #2: Yes: Andrea Spini

---

## [Author Response · Author response to Decision Letter 0]

28 May 2020

Reviewer #1: In this study authors carried out a validation study by using the Regional Healthcare administrative database of Umbria. As mentioned in the manuscript authors have published other article aimed to validate cardiovascular and cerebrovascular diseases. The article is well written, understandable and well structured. 

Thank you for your positive comment.

However, I have some comments that authors should take into account:

1) In the method section, the authors declare that they used ICD9 included in the primary position of the hospital discharge database. I am aware that sometime in second position can be included only a few information, but I was wondering why authors did not include also this codes in the validation (i.e., at least for most important event such as gastric haemorrhage).

Authors response: We decided to limit our validation study to only consider the codes in primary position because, according to the Italian legislation, the primary diagnosis constitutes the main cause of the need for treatment and/or diagnostic tests, and is mainly responsible for the use of resources.

We added this sentence to the study limitations: “We acknowledge that a potential limitation of our study is that we evaluated the accuracy of ICD-9 codes located only in primary position. We chose to limit our analysis only to the codes in primary position because, according to the Italian legislation, the primary diagnosis constitutes the main cause of the need for treatment and/or diagnostic tests, and is mainly responsible for the use of resources”.

2) In the discussion authors declare that other studies have been performed in Italy with the same purpose. Can authors better specify the difference among the other two studies and their one? In addition, can authors specify what their paper adds to the Italian literature on this topic?

Authors response: The main difference between our study and the other two Italian studies on this topic is that we were interested in validating the complete ICD-9 codes 531, 532, 534, and 578, while Cattaruzzi et al. and Pisa et al. considered the same ICD-9 codes of our study (peptic ulcer and gastrointestinal hemorrhage), but limited to the sub-codes of hemorrhage or perforation, with the aim to identify patients with upper gastrointestinal bleeding (UGIB) associated with intake of nonsteroidal antiinflammatory drugs (NSAIDs) and other drugs.

We decided to perform our study in order to validate for the first time the Regional administrative database of Umbria for these specific codes, that could be used in future to perform other epidemiological studies and health services research.

We added this in the discussion section: “The present study is one of the few in Italy and the first in Umbria Region validating the ICD-9 codes related to peptic ulcer and gastrointestinal hemorrhage using clinical charts as a reference standard”.

3) I partly agree with authors’s sentence “….validation studies of administrative database are context-specific, and thus our results can be applied only to the regional setting of Umbria”. What about the use/adaptation of the same codes in other Italian region? Do authors suppose a different diagnosis procedures/codes used in other Italian regions? If yes, why?

Authors response: In general, results that originate from a healthcare database are immediately applicable only to the setting in which such database has been validated, due to differences that may exist with respect to demographics, disease prevalence, and standards of care, among different regions. 

However, we think that our methodology could be replicated in other settings in order to identify possible differences in diagnostic accuracy results.

We amended the text as follows: “Regarding the generalizability of our study, we want to highlight that, in general, validation studies of administrative databases are context-specific due to differences that may exist in demographics, disease prevalence, and standards of care among different contexts, and thus our results can confidently be applied only to the regional setting of Umbria”.

Reviewer #2: Dear authors,

Your manuscript gives interesting informations on the accuracy of ICD-9 codes in identifying patients with peptic ulcer and gastrointestinal haemorrhage in the regional healthcare administrative database of Umbria and the validated codes will be useful for future epidemiological studies on health services. 

Thank you for your positive comment.

However, there are certain points that need to be clarified or better discussed:

1) In the "setting and datasource paragraph", no information on medical charts were presented. Please explain the origin of these medical charts. From which hospital did you retrive these data? Is there a data warehouse collecting these informations?

Authors response: We added in the text the description of the hospitals where the clinical charts came from: “We considered all the residents in the Umbria Region > 18 years discharged from seven hospitals (Perugia, Terni, Foligno, Città di Castello, Orvieto, Gubbio-Gualdo Tadino, Spoleto) between 2012 and 2014 with a diagnosis of peptic ulcer or gastrointestinal haemorrhage”.

We presented a minimal anonymized dataset containing the main characteristics of the study sample (see Supplementary material). 

2) Despite the great effort in analysing sensitivity and specificity of ICD-9codes, the authors measure those values only on the percentage of 80 “non cases”. Do you know the “real prevalence” of patients with these diseases in the Umbria region between 2012 and 2014? Is there a pathology registry in Umbria to compare the prevalence of diseases found using administrative data with the “real prevalence” of these diseases in an already validated registry? If these data cannot be recovered (or are inexistent) please discuss in the discussion section.

Authors response: Unfortunately, in Umbria there is not a pathology registry for ulcers or gastrointestinal hemorrhage, and therefore we do not know the real prevalence of the diseases. We based our estimation of sensitivity and specificity using a sample of “non-cases”, i.e. patients who had been discharged in the same period in a gastroenterology ward with a diagnosis other than peptic ulcer or gastrointestinal hemorrhage, from which we extracted a random sample of 80 patients.

We added in the discussion section the following sentence: “Instead, in order to estimate sensitivity and specificity, in absence of a disease registry for peptic ulcers and gastrointestinal hemorrhage that constitutes the real prevalence of the diseases, we chose to consider a sample of “non-cases”, i.e. patients who had been discharged in the same period in a gastroenterology ward with a diagnosis other than peptic ulcer or gastrointestinal hemorrhage, to individuate possible false negatives”.

3) Have you performed any sensitivity analyses considering not only ICD-9 CM codes in the primary position but also using the secondary positions? If not, please discuss further the motivations in using only primary position codes in the limitations of the study.

Authors response: We did not perform a sensitivity analysis considering the codes in secondary position because all our study sample was discharged with a diagnosis code in primary position (we did not extracted patients with the codes in secondary positions). 

We explained in the limitations the reasons why we chose to focus on the codes in primary position, adding the following sentence: “We chose to limit our analysis only to the codes in primary position because, according to the Italian legislation, the primary diagnosis constitutes the main cause of the need for treatment and/or diagnostic tests, and is mainly responsible for the use of resources”.

4) Please define abbreviations upon first appearance in the text. PPV and NPV abbreviation was firstly used in the "statistical analysis” section but the explanation is reported more than once in the “results” section.

Authors response: Thank you for your suggestion. We amended the text accordingly.

---

## [Editor Report · Decision Letter 1]

22 Jun 2020

Accuracy of ICD-9 codes in identifying patients with peptic ulcer and gastrointestinal hemorrhage in the regional healthcare administrative database of Umbria.

PONE-D-20-09372R1

Dear Dr. Abraha,

We’re pleased to inform you that your manuscript has been judged scientifically suitable for publication and will be formally accepted for publication once it meets all outstanding technical requirements.

Kind regards,

Gianni Virgili

Academic Editor

PLOS ONE
---

## [Editor Report · Acceptance letter]

24 Jun 2020

PONE-D-20-09372R1 

Accuracy of ICD-9 codes in identifying patients with peptic ulcer and gastrointestinal hemorrhage in the regional healthcare administrative database of Umbria. 

Dear Dr. Abraha:

I'm pleased to inform you that your manuscript has been deemed suitable for publication in PLOS ONE. Congratulations! Your manuscript is now with our production department. 

Kind regards, 

on behalf of

Dr. Gianni Virgili 

Academic Editor

PLOS ONE